# Design and Optimization of Flexible Polypyrrole/Bacterial Cellulose Conductive Nanocomposites Using Response Surface Methodology

**DOI:** 10.3390/polym11060960

**Published:** 2019-06-02

**Authors:** Yasong Chen, Fuying Wang, Lipan Dong, Zheng Li, Li Chen, Xinhai He, Jixian Gong, Jianfei Zhang, Qiujin Li

**Affiliations:** 1School of Material Science and Engineering, Tianjin Polytechnic University, Tianjin 300387, China; yasongchen@126.com (Y.C.); chenlis@tjpu.edu.cn (L.C.); 2School of Mathematical Science, Tianjin Polytechnic University, Tianjin 300387, China; 3Key Laboratory of Advanced Textile Composites (Tianjin Polytechnic University), Ministry of Education, Tianjin 300387, China; 15822955053@163.com (F.W.); 13072257279@163.com (L.D.); gongjixian@163.com (J.G.); zhangjianfei1960@126.com (J.Z.); vicmaldini@126.com (Q.L.); 4School of Textile Science and Engineering, Tianjin Polytechnic University, Tianjin 300387, China; 5School of Chemistry and Engineering, Tianjin University of Technology, Tianjin 300391, China; 6School of Physical Science and Technology, Tianjin Polytechnic University, Tianjin 300387, China; vmssci@163.com

**Keywords:** bacterial cellulose, pyrrole, response surface methodology, flexible conductive material

## Abstract

Flexible conductive materials have greatly promoted the rapid development of intelligent and wearable textiles. This article reports the design of flexible polypyrrole/bacterial cellulose (PPy/BC) conductive nanocomposites by in situ chemical polymerization. Box-Behnken response surface methodology has been applied to optimize the process. The effects of the pyrrole amount, the molar ratio of HCl to pyrrole and polymerization time on conductivity were investigated. A flexible PPy/BC nanocomposite was obtained with an outstanding electrical conductivity as high as 7.34 S cm^−1^. Morphological, thermal stability and electrochemical properties of the nanocomposite were also studied. The flexible PPy/BC composite with a core-sheath structure exhibited higher thermal stability than pure cellulose, possessed a high areal capacitance of 1001.26 mF cm^−2^ at the discharge current density of 1 mA cm^−2^, but its cycling stability could be further improved. The findings of this research demonstrate that the response surface methodology is one of the most effective approaches for optimizing the conditions of synthesis. It also indicates that the PPy/BC composite is a promising material for applications in intelligent and wearable textiles.

## 1. Introduction

Intelligent textiles are revolutionary fabrics based on the concept of bionics that integrate computing, digital components and electronics through novel technologies into textiles for advanced functionalities [1,2]. More concern is necessary for the fabrication of intelligent and wearable devices as they act as a part of the outfit when in touch with the human body [3]. The last decade has witnessed a rapid increase of interest in different fields including health monitoring, wearable gadgets, therapeutic devices with demanding properties such as flexible, light-weight, washable, bendable and conductive [4]. The increasing development of intelligent and wearable textiles has benefited from a major breakthrough in emerging flexible and conductive materials, which enable a flexible combination of conductive materials and traditional non-conductive textiles. 

Although conventional metal conductors possess high conductivity, they cannot meet the requirements of flexible electronic equipment for elastic stretching and bending of materials [5], such as supercapacitors, actuators, sensors. At present, electronically conducting polymers (EPCs) including polypyrrole (PPy), polythiophene (PTh) and polyaniline (PANI) are typically used to coat on cellulose substrates for composite materials. They have excellent conductivity, controllable synthesis process and mechanical flexibility [6,7,8,9,10]. Among the promising EPCs, PPy and its derivatives have received great attention due to their high conductivity, high energy density, good environmental stability and low toxicity [10,11,12]. Deposition of PPy on a fiber surface of fabrics and yarn, such as silk [13], cotton [14,15], polyester [16] and polyamide [17] has been widely investigated. These conductive materials were generally obtained through in situ oxidative polymerization of pyrrole in the presence of textiles [18]. 

Among polymer fibers, bacterial cellulose (BC) is one kind of attractive biomaterial that shows outstanding properties together with a completely biodegradable and biocompatible nature [19]. BC is an unbranched polysaccharide, comprising linear chains of β-1,4-glucopyranose residues produced by microorganisms belonging to *Acetobacter xylinum* [20]. It possesses high crystallinity, purity and higher flexibility than plant cellulose or other cellulose derivatives [21,22,23,24]. Its chemical structure with the presence of hydroxyl and ether groups provides an excellent hydrophilic matrix for nanoparticles incorporation [25]. There have been some reports in the literature about the in situ production of PPy nanoparticles on the BC surface [26,27,28,29,30,31,32]. However, due to the aggregation of PPy nanoparticles on BC fibers under un-optimized synthesis conditions, the electrical conductivities of the composites were relatively low (2.1–6.8 × 10^−2^ S cm^−1^) [33], which limited their applications in emerging technologies. In general, many factors such as pyrrole amount, polymerization time, molar ratio of oxidant (or dopant) to pyrrole and temperature have direct effects on the conductivity of PPy/BC composite. To our best knowledge, no attempts have been made on process optimization and interactions of two different factors were rarely carried out.

Based on the classical method, in order to conduct such experiments, one needs to change one parameter at each time and maintain other parameters constant. A great number of experiments might be performed which spend lots of time and energy. In order to reduce the limitations of the classical method, collective optimization of all the effective parameters via an empirical design based on statistics could be employed in the optimization process, like RSM which stands for response surface methodology [34]. Box-Behnken response surface methodology is a powerful experimental design tool and has been used to optimize and understand the performance of complex systems [35,36,37,38]. This design reduces the number of experimental runs which saves time and money. It does not contain factor level combinations where the factors are all at their higher or lower levels, they may be useful in avoiding experiments under extreme conditions, for which unsatisfactory results may occur [39]. 

In this study, a kind of flexible and conductive nanocomposite was prepared by in situ chemical polymerization of pyrrole using bacterial cellulose as a three-dimensional template. Box-Behnken design (BBD) was applied for the RSM in the experimental design with three independent variables at three levels. The three independent variables considered in this study were pyrrole amount, the molar ratio of hydrochloric acid (HCl) to pyrrole and polymerization time. The aim of the current research was to gain the synthesis conditions which were optimum for electrical conductivity through an analysis of impacts of three important factors of the operation. Moreover, thermal, electrochemical and morphological properties of the prepared PPy/BC composite were evaluated. We believe that our work would contribute to the progress in flexible and conductive materials.

## 2. Materials and Methods

### 2.1. Materials

BC pellicles were supplied by Hainan Yide Food Co., Ltd., Haikou, China. Pyrrole was purchased from Shanghai Kefeng Industrial Co., Ltd., Shanghai, China. HCl and ferric chloride hexahydrate (FeCl_3_·6H_2_O) were obtained from Tianjin Sailboat Chemical Reagent Technology Co., Ltd., Tianjin, China. All the chemical reagents were used as received without further purification.

### 2.2. Preparation of PPy/BC Composite by in Situ Chemical Polymerization

The preparation procedure of PPy/BC composites by in situ chemical polymerization was described in the previous work [19,28,40]. In this paper, a kind of flexible conductive material was designed using BC fiber suspension as a template. Firstly, the BC membranes were purified by being washed in distilled water several times. Then, they were soaked with distilled water for 12 h to fully swell and gently wiped off any excess moisture on the surface by a clean gauze. In order to prepare 1000 mL BC fiber suspension, the BC membrane with a mass of 98.68 g was cut into small cubes, then broken and dispersed with JZPB-165J01 Soymilk Machine (Zhongshan Jinzheng Life Electric Appliance Co., Ltd., Zhongshan, China) for 5 min. The fiber suspension was diluted into 1.5 mg mL^−1^ by adding some distilled water. The suspension (50 mL) was added into the solution of HCl (2 M) and a certain amount of ferric chloride (FeCl_3_, the molar ratio of FeCl_3_ to pyrrole was fixed at 1.38). Subsequently, the mixture was sonicated for 15 min at a power of 100 W, magnetic stirred for 15 min and cooled to 4 °C; a certain amount of pyrrole monomer was added into the above mixture and kept stirring at 4 °C for some time to obtain PPy/BC composite slurry (the molar ratio of FeCl_3_ to pyrrole was fixed at 1.38). The slurry was vacuum filtered to form a film, which was then freezing dried at −40 °C for 24 h. Finally, the PPy/BC composite was finished.

### 2.3. Box-Behnken Design

In the analysis of multi-factor experiments, the regression relationship presents a curve or a curved surface between the test index and multiple test factors, called response surface methodology (RSM). In this paper, RSM was employed for optimization of process parameters of PPy/BC composites using software Design Expert v.8.0.5b. The Box-Behnken design (BBD) was applied for the RSM in the experimental design with three independent variables at three levels. The three independent variables considered in this study were pyrrole amount (X_1_), molar ratio of hydrochloric acid to pyrrole (X_2_) and polymerization time (X_3_). 

Due to the different standards of each variable, it was necessary to normalize them prior to the analysis of the regression. The levels of variables (low, medium and high) were recorded as −1, 0, and +1, respectively. A linear transformation was applied in order to code the actual variables ($X_{i}$) shown in Equation (1).
(1)$$x_{i}=\frac{X_{i}-\left(X_{\max}{+X}_{\min}\right)/2}{\left(X_{\max}-X_{\min}\right)/2}$$
where $x_{i}$ was the coded value of the variable, $X_{i}$ was the actual value of variable, $X_{\max}$ stood for the maximum actual value of the variable and $X_{\min}$ represented the minimum actual value of the variable. It was obvious that “$\left(X_{\max}{+X}_{\min}\right)/2$” referred to the actual value of $X_{i}$ at the center point and “$X_{\max}-X_{\min}$” corresponded to the change of value. Table 1 presentes the experimental range and levels of independent variables of design and preparation PPy-BC nanocomposites. 

The experimental levels for each variable were selected based on the results of preliminary experiments and references [19,27,41]. A total of 17 experimental runs with different combinations of three factors were carried out. A full quadratic model was fitted to correlate the relationship between independent variables and response value represented by Equation (2).
(2)$${Y=\beta }_{0}+\overset{k}{\underset{i=1}{\sum }}\beta_{i}X_{i}+\overset{k}{\underset{i=1}{\sum }}\beta_{\mathrm{ii}}X_{i}^{2}+\overset{k}{\underset{i=1}{\sum }}\overset{k}{\underset{i<j}{\sum }}\beta_{\mathrm{ij}}X_{i}X_{j}+\varepsilon$$
where Y was the response value of electrical conductivity, X_i_ and X_j_ were the value of independent variables (i and j were used to index the range of the pattern from 1 to k), k represented the number of independent variables (k = 3), β_0_ was a constant, β_i_ stood for the linear coefficient, β_ii_ referred to the second order effect on regression coefficients, β_ij_ corresponded to the interaction coefficient, and ε was the statistical error of the model [34]. The response value Y was regarded as the comprehensive impact of the various factors. If the coefficient value before the item was more than zero, it indicated that the item would have a positive effect on response; if the coefficient value before the item was less than zero, the response value would decrease as the item increased.

### 2.4. Characterization and Electrical Measurements

A double-electric measurement digital tester with four-point probe (ST2263, Suzhou Jingge Electronics Co., Ltd., Suzhou, China) was used to assess the square resistance of PPy/BC composite (with a size of 2 cm × 2 cm) at room temperature. Its thickness was averaged three times by Thickness Gauge (Taizhou Ai Test Instrument Co., Ltd, Taizhou, China.). And the conductivity could be calculated according to Equation (3).
(3)C=1/(R×d)
where C (S cm^−1^), R (Ω), d (cm) were the electrical conductivity, square resistance and thinness of the composite, respectively.

The morphology of the samples was characterized by field emission scanning electron microscope (FE-SEM, Hitachi S-4800, Tokyo, Japan). 

The FT-IR spectra were recorded by Frontier Infrared Spectrometer (Thermo Fisher Scientific) over a wave number range from 420 to 4000 cm^−1^.

A thermos gravimetric analyzer was used to study its thermal stability with a heating rate of 10 °C min^−1^ in a nitrogen atmosphere.

Cyclic voltammetry (CV), galvanostatic charge−discharge (GCD) and electrochemical impedance spectroscopy (EIS) were carried out at room temperature with CHI Electrochemical Analyzer/Work Station (Shanghai Chenhua Instrument Co., Ltd., Shanghai, China). A piece of PPy/BC composite (with a size of 1 cm × 0.5 cm) was immersed in 2.0 M LiCl electrolyte and used as the working electrode. No binders or other conductive materials were added. A titanium plate and an Ag/AgCl were used as a counter electrode and a reference electrode, respectively. In the three-electrode system, CV and GCD were performed in the potential range from −0.2 V to 0.6 V. EIS was recorded in the frequency range from 0.01 Hz to 100 kHz with an amplitude of 5 mV. The areal capacitance could be obtained by Equation (4) [42].
(4)$$C_{s}=\frac{I\times \Delta t}{S\times \Delta V}$$
where Cs (F cm^−2^), I (A), Δt (s), S (cm^2)^, ΔV (V) were the areal capacitance, the discharge current, the discharge time, area and the potential window, respectively.

## 3. Results and Discussion

### 3.1. Synthesis of PPy/BC Composite

Figure 1 illustrates the fabrication process of PPy/BC composites. Firstly, the desired amount of HCl and FeCl_3_ solution was mixed with BC nanofibers suspension and ultrasonic cleaning for 15 min to ensure that it was mixed well. Then, the pyrrole monomer was added into the solution after the temperature was cooled down to 4 °C. It was clear to see that the color of BC nanofibers suspension changed from white to black gradually due to polymerization of pyrrole in the presence of FeCl_3_. Finally, the mixture was vacuum filtered through a filtration membrane, followed by freeze-drying to gain the paper-like flexible PPy/BC composite. In contrast to the previous method [43], we intentionally omitted the elution steps for more HCl retention. The composites obtained exhibited the conductivities in the range from 2.52 to 7.81 S cm^−1^ (see Table 2 for details), which were much higher than those of PPy/BC composites after being rinsed several times with deionized water (up to 2.26 S cm^−1^).

Figure 2 describes the formation mechanism of PPy chain. FeCl_3_ acted as the oxidant creating chemically active cation-radicals of the monomers. The coupling between two radicals resulted in the formation of a bond between their α-positions and the formation of dihydromer dication. The loss of two protons formed the aromatic dimer. Polymer chains continued to grow as long as pyrrole and FeCl_3_ were available [44]. PPy chain had a perfect ultra-long conjugated structure with C=C bonds and C-C bonds alternately arranged. If the molar ratio of FeCl_3_ to pyrrole was high, more counter anions would be left in the solution at the completion of the polymerization reaction. If the molar ratio of FeCl_3_ to pyrrole was low, the monomer may not be completely polymerized and conjugation lengths of PPy formed would be diminished [45]. The PPy chain was simultaneously doped by HCl during polymerization. The combination of the protons of the dopant and the carbon atoms on PPy backbone caused a reduction in the density of electron clouds on the carbon atoms for electrical conduction. The reaction mechanism of the formation of PPy was also discussed by Sadki et al. [46]. 

Figure 3 presents the scanning electron microscope (SEM)micrographs of the pure BC membrane and PPy/BC nanocomposite. The BC membrane consisted of continuous nanofibers with a diameter of about 60 nm. A smooth surface morphology and three—dimensional porous network structure among the BC nanofibers could be clearly identified (Figure 3a). After polymerization, it was found that PPy nanoparticles deposited on the fiber surface connected to form a continuous nanosheath structure with a diameter of around 160 nm by taking along BC nanofibers (Figure 3b). The hydrogen bonds between imine groups of pyrrole and hydroxyl groups of BC might serve as a traction force to assist the growth of the continuous nanosheath of PPy over cellulose and avoid the large-scale aggregate formation [47,48]. This demonstrated that BC nanofiber could work as a good template for the polymerization of pyrrole for highly conductive composite [25].

### 3.2. Response Surface Methodology Analysis

A set of 17 experiments was run according to a 3^3^ Box-Behnken design. The results are shown in Table 2. The quadratic model in terms of coded factors for the response is presented in Equation (5). While a positive sign before a term showed a synergistic impact, a negative sign represented an antagonistic one [49].
(5)$$Y=7.14+0.29X_{1}+0.22X_{2}-0.36X_{3}+0.24X_{1}X_{2}-0.85X_{1}X_{3}+0.21X_{2}X_{3}-2.26X_{1}^{2}-1.15X_{2}^{2}-0.83X_{3}^{2}.$$

Equation (4) indicates that the pyrrole amount, which had a coefficient of 0.29, had the biggest positive linear impact among other parameters [34]. The interaction between pyrrole amount and polymerization time with a coefficient of 0.85 had the most negative impact on the conductivity of PPy/BC composite.

The results of the second-order response surface model in the form of analysis of variance (ANOVA) are shown in Table 3. The statistical significance of the model equation was evaluated by F-values and *p*-values. The F-value and *p*-value of the model were 42.68 and 0.0007, respectively, which explained the stability and significance of the model [50]. In the same way, $X_{1}$, $X_{2}$, $X_{3}$, $X_{1}X_{2}$, $X_{1}X_{3}$, $X_{2}X_{3}$, $X_{1}^{2}$, $X_{2}^{2}$ and $X_{3}^{2}$ were also reported. The decision coefficient R^2^ explained the variability in the response values by the experimental parameters and their interactions [51]. R^2^ was calculated to be 0.9543, which implied 95.43% of the variation could be explained by this model and only 4.57% were not [52]. Adjusted R^2^, a measure of fit, was determined to be 0.8955. The difference between decision coefficient R^2^ and adj-R^2^ was logically acceptable [53,54]. Both R^2^ and adj-R^2^ were higher than 0.80, which suggested that the model had a high efficiency for the representation of the experimental data [55]. The non-significant value of lack of fit (more than 0.05) showed that the quadratic model was valid for the present study [56,57]. Thus, this model was suitable for determining the optimum condition of the PPy/BC nanocomposites preparation.

#### 3.2.1. Effect of Pyrrole Amount and Molar Ratio HCl to Pyrrole on Conductivity

The response surface plot for electrical conductivity of PPy/BC nanocomposites for pyrrole amounts ranging from 0.1 mL to 0.5 mL and the molar ratio of HCl to pyrrole ranging from 5 to 35 against polymerization time of 180 min is shown in Figure 4a. The electrical conductivity initially increased and then decreased with a further increase in pyrrole amount. This was mainly due to more PPy formation at higher monomer concentration, but too much pyrrole might cause circuit blockage to affect electron conduction. As the molar ratio of HCl to pyrrole increased, conductivity initially decreased and then increased. HCl functioned as the dopant causing defects in PPy backbone and leading to the decrease of electron cloud density. With a further increase in dopant, there might be a reduction in intermolecular hopping of charge carriers due to an increased gap between the polymer chains by bulky side groups of the anions and thereby adversely affecting conductivity [58]. The drop might reflect that the detachment of PPy from the BC surface had occurred [59].

#### 3.2.2. Effect of Pyrrole Amount and Polymerization Time on Conductivity

The response surface plot for electrical conductivity of PPy/BC nanocomposites for pyrrole amount ranging from 0.1 mL to 0.5 mL and polymerization time ranging from 60 min to 300 min against the molar ratio of HCl to pyrrole of 20 is shown in Figure 4b. It could be seen clearly that with the increase pyrrole amount the conductivity increased initially and then decreased. As the polymerization time increased, the conductivity first increased and then decreased. The increase in conductivity might be attributed to more polymer formation and uniform polymer deposition due to more time available, while the decrease at high polymerization time might be ascribed to the over-oxidation of the polymer leading to the deterioration of the electrical properties [60]. Similar observations were reported by Maiti et al. [61].

#### 3.2.3. Effect of Molar Ratio of HCl to Pyrrole and Polymerization Time on Conductivity

The response surface plot for electrical conductivity of PPy/BC nanocomposites for the molar ratio of HCl to pyrrole ranging from 5 to 35 and polymerization time ranging from 60 min to 300 min against pyrrole amount of 0.30 mL is shown in Figure 4c. As expected, a similar trend of the molar ratio of HCl to pyrrole and polymerization time was observed here. The electrical conductivity initially increased and then decreased with the further increase in the molar ratio of HCl to pyrrole. As the polymerization time increased, the conductivity first increased and then decreased.

The optimal preparation process of PPy/BC nanocomposite by Design-Expert. 8.05b was as follows: the pyrrole amount was 0.32 mL, the molar ratio of HCl to pyrrole was 20.94:1, polymerization time was 147.81 min and the predicted maximum conductivity was 7.21 S cm^−1^. In the actual operation process, the preparation parameters were simplified to 0.32 mL of pyrrole amount with a molar ratio of HCl to the pyrrole of (21:1) and a polymerization time of 150 min. After three verification tests, the conductivity obtained was (7.34 ± 0.25) S cm^−1^, which was only 1.8% different from the predicted value, indicating that the regression model obtained by the response surface method was reliable and effective. Additionally, the electrical conductivity in this study was much higher than those of other PPy/cellulose fibers (1.9 × 10^−2^–6.63 S cm^−1^) [25]. Compared with the conductivities 1.14 S cm^−1^ for pure PPy [62], PPy/BC nanocomposites gave a 6-fold increase in conductivity, maybe due to better doping and more porous structures.

### 3.3. Fourier Transform Infrared Spectroscopy

Figure 5 presents the Fourier transform infrared (FT-IR) spectra of pure BC and PPy/BC membranes. The spectrum of pure BC membrane showed strong signals at 3334 cm^−1^, 2884 cm^−1^ and 1017 cm^−1^ respectively. These signals corresponded to the characteristic absorption peak of BC. The absorption peak at 3334 cm^−1^ was attributed to the intermolecular hydrogen bond stretching vibration caused by O–H groups. The absorption peak at 2884 cm^−1^ was produced by asymmetrically stretching vibration of C–H [63]. The peak at 1017 cm^−1^ was caused by the stretching vibration of C–O bonds. Compared to BC, a group of characteristic bands of PPy could be indexed in the PPy/BC composite, suggesting a successful polymerization [64]. The peaks at 1007 cm^−1^, 1131 cm^−1^ and 1516 cm^−1^ were attributed to C–H out-of-plane deformation vibration, C=C and C–C in-ring-stretching, respectively [19]. The C–N stretching vibration peaks of PPy appeared at 1288 cm^−1^ and 1435 cm^−1^ [42].

### 3.4. Thermal Stability Analysis

The thermal stabilities of pure BC membrane and PPy/BC nanocomposite were investigated by thermogravimetric analysis (TGA) and derivative thermogravimetric (DTG) depicted in Figure 6. The pure BC membrane experienced mass losses were observed in three stages. From room temperature to 100 °C, 3.64% weight loss could be seen due to the hydration of water absorbed physically or bound by hydrogen bond on BC. The second weight loss at a higher temperature, which took place at 300 °C, was attributed to the decomposition of the amorphous BC region. The third stage at around 450 °C corresponded to the decomposition polymeric chains. For PPy/BC composite, the second stage of weight loss was a continuous process. The composite decomposed at a lower speed than pure cellulose. As shown in Figure 6b, at the second peak, the speed of weight loss shifted from 22.04% min^−1^ for cellulose to 2.32% min^−1^ for the composites. The PPy layer was the most external and more stable component that acted as a protective barrier against BC thermal degradation [18]. As a result, the composites exhibited higher thermal stability than pure cellulose. A similar result was obtained for PPy-coated silk fabrics [65], which was attributed to the interaction between cellulose and PPy.

### 3.5. Electrochemical Performances

In order to investigate the electrochemical performances of the PPy/BC nanocomposite prepared in optimal conditions, it was immersed in a 2.0 M LiCl electrolyte with a three-electrode configuration and explored by cyclic voltammetry (CV), galvanostatic charge-discharge (GCD), electrochemical impedance spectroscopy (EIS) measurements.

As shown in Figure 7a, the CV curves at different scan rates were nearly rectangular and basically symmetrical about the zero, demonstrating the ideal capacitive behavior. Moreover, as the scan rate increased, the shape of the curves was hardly changed. Figure 7b presented the GCD curves at different current densities from 1 to 10 mA cm^−2^. They were nearly linear and symmetrical without an obvious *IR* drop, suggesting a fast and good reversibility charge-discharge property of the electrode material [66]. At the discharge current density of 1 mA cm^−2^, the highest areal capacitance could reach 1001.26 mF cm^−2^. This value was higher than that of PANI network/Au/paper electrode (0.8 F cm^−2^) [67], PANI@carbon and PPy@carbon (787.40 mF cm^−2^ and 136.99 mF cm^−2^) [68] and others (0.4–30 mF cm^−2^) reported in literature at the same discharge current density [69,70,71,72]. Simultaneously, it was found that the PPy/BC electrode retained about 90.97% of the initial capacitance after 80 cycles (Figure 7c). It should be pointed out that the cycling stability needed to be improved for practical applications. Figure 7d showed the Nyquist plot of PPy/BC nanocomposite, which was made up of a semicircle at high-frequency region followed by a straight line at low-frequency region [43]. The diameter of the semicircle in the high to the midfrequency region represented the electron transfer resistance at the interface of the electrodes and electrolytes. The vertical line in the low-frequency region was caused by the charges stored in the electrochemical reaction. The slope of the line was indicative of the perfect capacitive behavior. It was found that the electrode had a low internal resistance, while the vertical line deviated from the imaginary axis indicating a slightly poor tantalum capacitance. 

## 4. Conclusions

Based on the response surface methodology, a kind of flexible and conductive PPy/BC nanocomposite was successfully prepared by in situ chemical polymerization of pyrrole in the presence of BC. A maximum electrical conductivity of 7.21 S cm^−1^ was obtained by optimizing the process at a pyrrole amount of 0.32 mL, the molar ratio of HCl to pyrrole of (21:1) and polymerization time of 150 min. In the suggested optimal conditions, the experimentally determined response of (7.34 ± 0.25) S cm^−1^ was reasonably close to the predicted response. Moreover, the composite exhibited a high areal capacitance of 1001.26 mF cm^−2^ at the discharge current density of 1 mA cm^−2^, but its cycling stability could be further improved. The finding of this research demonstrates that RSM based on BBD is one of the most effective approaches for optimizing the conditions of synthesis. It also provides insight for the use of PPy/BC nanocomposites as flexible and conductive materials for intelligent and wearable textiles.

## Figures and Tables

**Figure 1 polymers-11-00960-f001:**
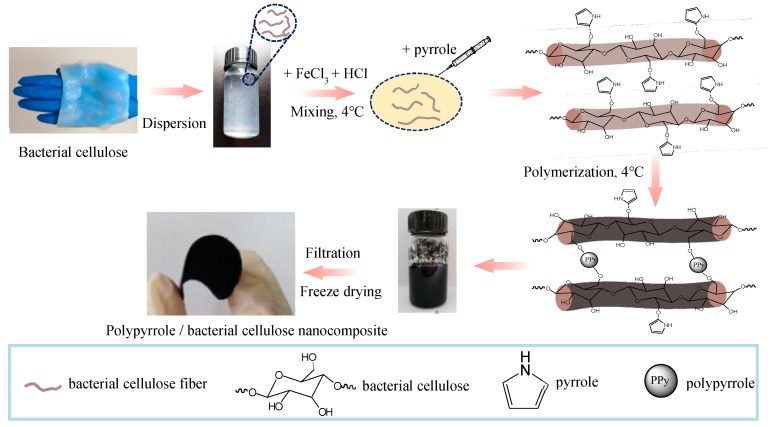
Schematic illustration of preparation PPy/BC nanocomposite.

**Figure 2 polymers-11-00960-f002:**
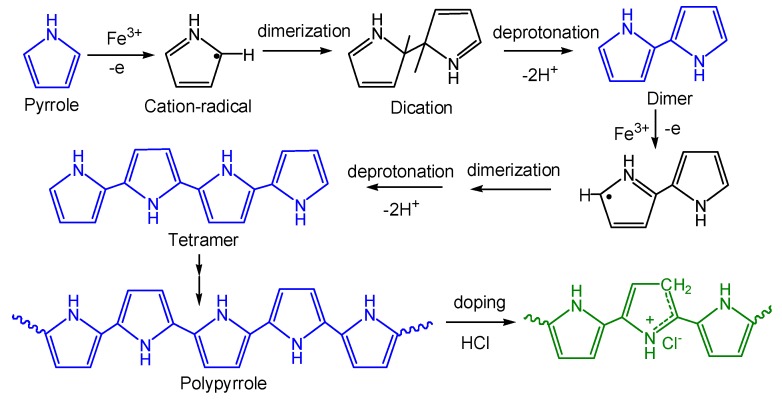
Schematic diagram of the polypyrrole formation mechanism.

**Figure 3 polymers-11-00960-f003:**
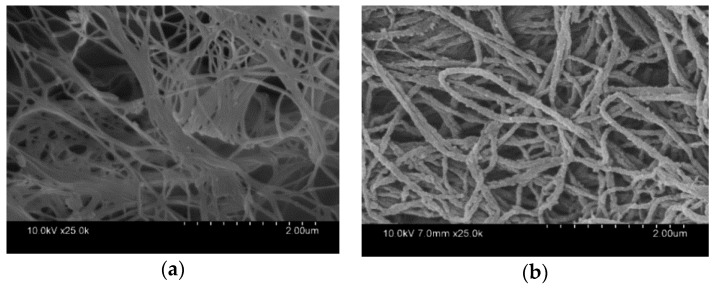
Scanning electron microscope (SEM) images of pure bacterial cellulose (BC) and PPy/BC composite: (**a**) pure BC membrane; (**b**) PPy/BC composite.

**Figure 4 polymers-11-00960-f004:**
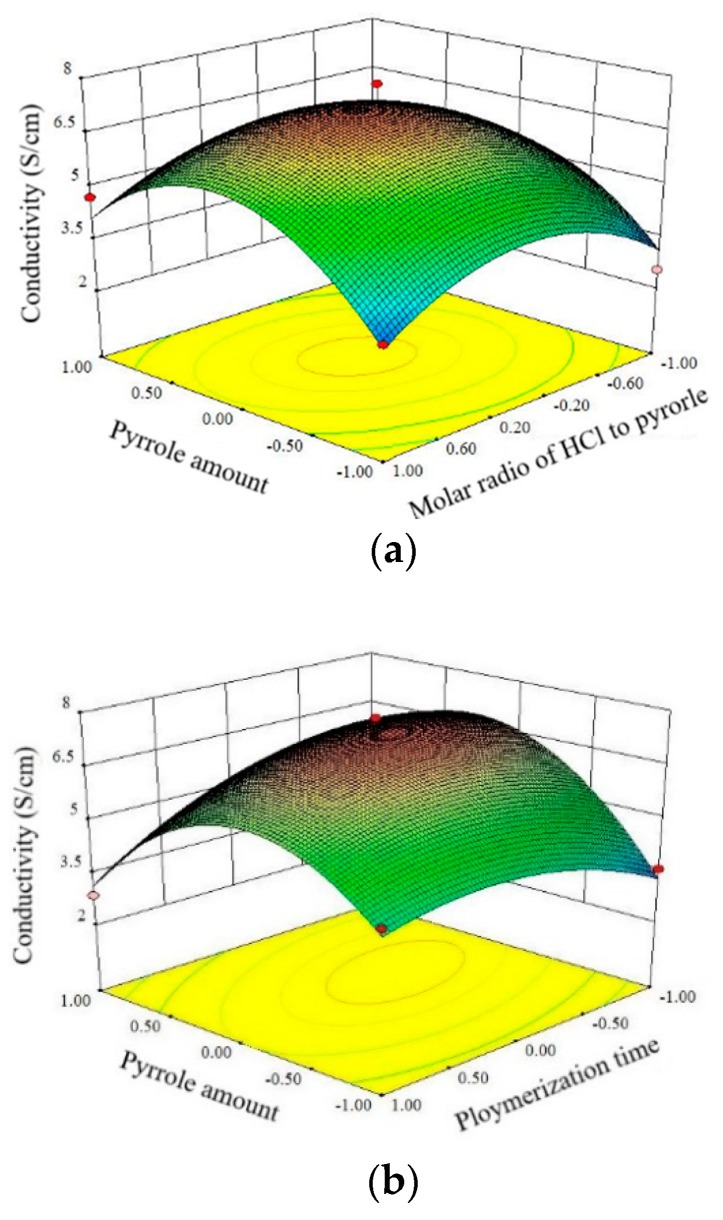
Response surface plots of independent variables on the conductivity of PPy/BC composite: (**a**) pyrrole amount and the molar ratio of HCl to pyrrole; (**b**) pyrrole amount and polymerization time; (**c**) molar ratio of HCl to pyrrole and polymerization time.

**Figure 5 polymers-11-00960-f005:**
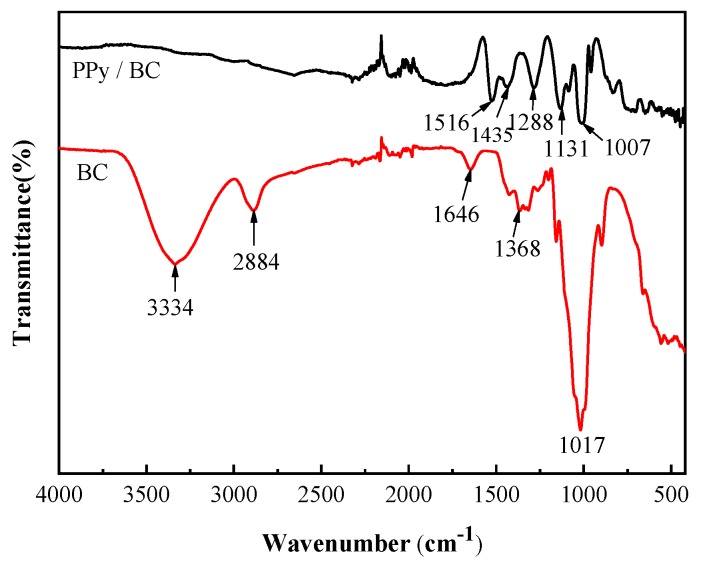
Fourier transform infrared (FT-IR) spectra of pure BC and PPy/BC membranes.

**Figure 6 polymers-11-00960-f006:**
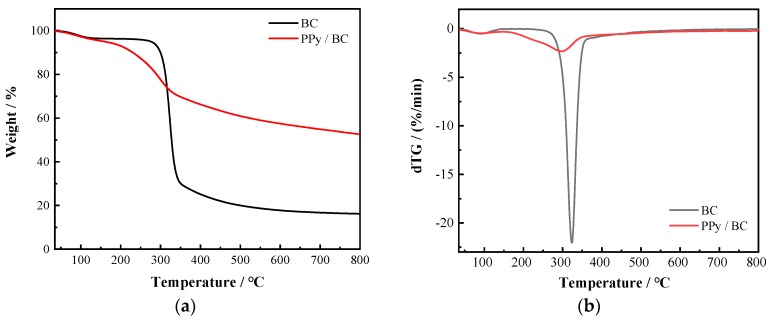
(**a**) Thermogravimetric analysis (TGA) and (**b**) derivative thermogravimetric (DTG) curves of pure BC and PPy/BC nanocomposite.

**Figure 7 polymers-11-00960-f007:**
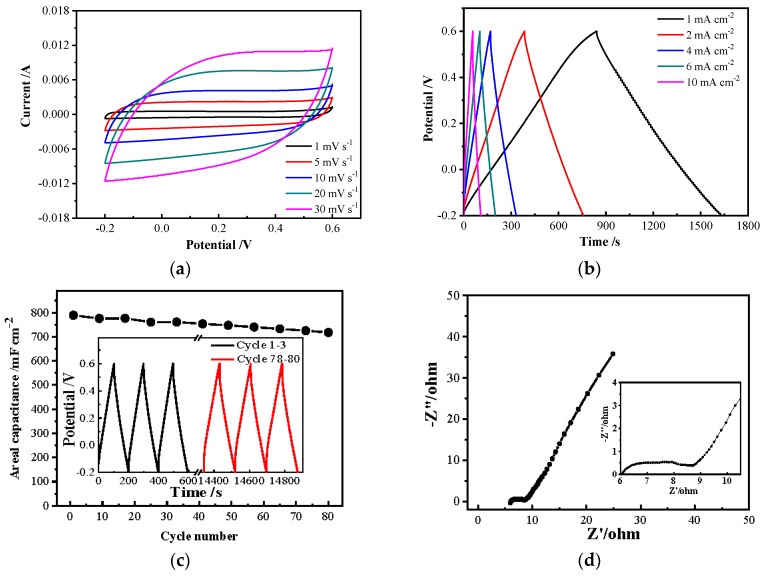
Electrochemical performances of PPy/BC nanocomposite prepared based on response surface: (**a**) cyclic voltammetry curves at different scan rates; (**b**) galvanostatic charge-discharge curves at different current densities; (**c**) cycle stability; (**d**) electrochemical impedance spectroscopy curves.

**Table 1 polymers-11-00960-t001:** Experimental range and levels of independent variables.

Factors	Labels	Levels		
		−1	0	1
Pyrrole amount/mL	X_1_	0.1	0.3	0.5
Molar ratio of HCl to pyrrole	X_2_	5	20	35
Polymerization time/min	X_3_	60	180	300

**Table 2 polymers-11-00960-t002:** Response surface design matrix with experimental and predicted values.

Run	X_1_	X_2_	X_3_	Y Conductivity/(S cm^−1^)		Error/(S cm^−1^)
				Experimental	Predicted	
1	0	1	−1	4.78	5.17	−0.39
2	0	0	0	6.93	7.14	−0.21
3	0	0	0	7.03	7.14	−0.11
4	1	−1	0	3.08	3.20	−0.12
5	−1	0	−1	3.54	3.27	0.27
6	0	−1	−1	5.48	5.16	0.32
7	−1	0	1	4.47	4.27	0.20
8	1	1	0	4.71	4.12	0.59
9	0	−1	1	4.41	4.02	0.39
10	0	0	0	7.08	7.14	−0.06
11	−1	−1	0	2.52	3.11	−0.59
12	0	0	0	6.87	7.14	−0.27
13	0	0	0	7.81	7.14	0.67
14	1	0	1	2.86	3.13	−0.27
15	1	0	−1	5.35	5.55	−0.20
16	−1	1	0	3.18	3.06	0.12
17	0	1	1	4.56	4.88	−0.32

**Table 3 polymers-11-00960-t003:** Analysis of variance (ANOVA) of response surface quadratic model.

Source	Sum of Squares	Degree of Freedom	Mean Square	F-Value	*p*-Value Prob > F	Significance
Model	42.68	9	4.74	16.23	0.0007	**
$X_{1}$	0.66	1	0.66	2.24	0.1779	
$X_{2}$	0.38	1	0.38	1.3	0.2925	
$X_{3}$	1.02	1	1.02	3.47	0.1046	
$X_{1}X_{2}$	0.24	1	0.24	0.81	0.3994	
$X_{1}X_{3}$	2.92	1	2.92	10.01	0.0159	*
$X_{2}X_{3}$	0.18	1	0.18	0.62	0.4575	
$X_{1}^{2}$	21.54	1	21.54	73.73	<0.0001	**
$X_{2}^{2}$	9.59	1	9.59	32.83	0.0007	**
$X_{3}^{2}$	2.88	1	2.88	9.86	0.0164	*
Residual	2.05	7	0.29			
Lack of Fit	1.46	3	0.49	3.36	0.1363	
Pure Error	0.58	4	0.15			
Cor Total	44.73	16				
$R^{2}$ = 0.9543 $R_{\mathrm{Adj}}^{2}$ = 0.8955						

Note: values of “Prob > F” less than 0.0500 indicated model terms were significant (*) and less than 0.0100 were extremely significant (**).

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
