# Peer review of "Design and Optimization of Flexible Polypyrrole/Bacterial Cellulose Conductive Nanocomposites Using Response Surface Methodology"

_polymers, 2019, doi:10.3390/polym11060960_

Reviewer 1 Report

The work presents the detailed, considered and scrupulous investigation fulfilled with assistance of modern mathematician, statistic and instrumental methods utilizing modern equipment. It is written in good English.

At the same time I have some remarks and questions.

1)      Maybe it is better to use “pyrrole amount or pyrrole portion” instead of “pyrrole dose”;

2)      Line 59:  to use “the presence of hydroxyl and ether groups” instead of “the presence of hydroxyl and ether bonds”;

3)      Why it is necessary to wipe off any excess of moisture if afterwards the suspension of a sample (it is not clear in what solution the suspension was done) was added into the water solution of HCl (2M); if it is necessary to wipe off the rest of impurities or the dried sample could be ground better, then it is necessary to explain here, in the description of the procedure.

4)      What is the role of ferric chloride and what was the amount of FeCl3+;

5)      Lines 95-106

“In order to prepare BC fiber suspension (in what???, what is the amount of BC?), the BC membrane was cut into small cubes, then broken and dispersed with JZPB-165J01 Soymilk Machine (Zhongshan Jinzheng Life Electric Appliance Co., Ltd.) for 5minutes. As described previously, the fiber suspension (in what???, was diluted into 1.5 mg mL-1. 50 mL suspension was added into the solution of HCl (2 M) and ferric chloride (FeCl3) (what amount of this reagent?)”

6)      Lines 149-150 Results and Discussion

I have many questions to the Scheme depicted in Figure 1. Once again – what is the role of ferric chloride?

a)      First of all, pyrrole can be polymerized with the help of mineral acids (catalytic amounts) in aqueous conditions,

b)      at the same time, it is well known that poly(pyrrole) is usually prepared with the help of anhydrous ferric chloride in organic media (nitromethane, chloroform, etc.).

c)      nevertheless, there are the works concerning polymerization of pyrrole in aqueous media in presence of Fe(III) compounds; e.g., Bjorklund [10.1039/F19878301507] reported the kinetics of chemical oxidation of pyrrole by FeCl3 in aqueous solutions of methylcellulose. He monitored the decrease of pyrrole by measuring the absorption at 800 nm, and found that the kinetics curves exhibited behavior characteristic of autocatalytic reactions.

The proposed mechanism is also dependent on methylcellulose. D.-V. Brezoi [J. Sci. Arts, 2010, 1 (12), 53-58] also described mechanism of pyrrole polymerization in aqueous media by oxidation with the help of ferric chloride.

The same was investigated by D. Beneventi et al. (0.1007/s10570-006-9077-9). They explain the process of polymerization as follows: «Iron is adsorbed on cellulose fibres in FeCl3 solutions as multivalent ferric compounds, thereby inducing fibre cationization; nevertheless, the amount of Fe3+ adsorbed/retained on fibres with ferric chloride concentrations lower than approximately 0.5 mol/l is not sufficient to promote pyrrole polymerization”

d)      Here, in the Manuscript under discussion, one can see, that the authors have supposed formation of chemical bonds between pyrrole and BC (Figure 1), as the next step they have shown the result of polymerization as they have supposed, but the process of polymerization has not been explained. As they used a large amount of HCl (HCl : pyrrole = 21:1) then polymerization can be performed by the action of HCl.  But! It is necessary to remember, that pyrrole is very sensitive to acids, it has a high acidofuge factor. 

e)      So, as to me it would be better to explain the role of HCl, FeCl3, amount of FeCl3, to clarify amount of BC sample (in order to understand the proportions between BC, pyrrole, HCl and FeCl3) and to let a reader have the details of mechanism, otherwise it is not clear what is the nature of a pyrrole polymer prepared. As to me the section 3.2.1. only describes the results but does not explain them. What is the structure of the “ultralong conjugated PPy chains”? It would be better to determine this term.

So this work could be published in this Journal. But it seems to me that it would be better to clarify the problems I have written about..

Author Response

    We appreciate you very much for positive and constructive comments and suggestions on our manuscript entitled “Design and Optimization of Flexible Polypyrrole / Bacterial Cellulose Conductive Nanocomposites Using Response Surface Methodology”.

    We have studied your comments carefully and have made revision which marked in red in the paper. We have tried our best to revise our manuscript according to the comments. 

    Thank you and best regards.

Reviewer 2 Report

In this paper, Chen et al. report some interesting properties of conductive nano composites obtained by means of "in situ" chemical polymerisation.

Authors investigate the effect of pyrrole dose, HCl concentration and polymerisation time on the behaviour of polypyrrole-bacterial cellulose composites. Investigation was based on different experimental techniques such as electrical conductivity measurements, field emission scanning electron microscopy, FT-IR spectroscopy, gravimetric analysis and electrochemical impedance spectroscopy.

The experimental pert is appropriately carried out and I have no criticisms to do.However, before the manuscript can be considered for publication, I believe that the authors should consider the following two points.

First, the statistical mode they use. Here, eq. 1 must be commented a little more in detail since not all readers are familiar with this type of analysis.

Second, the impedance spectroscopy curve shown in Fig. 6d (Nyquist plot) evidences the presence of multiple relaxation processes. Authors should at lest give some indication of the nature of these processes. Authors say only that the diameter of the semicircle represents the resistance at the electrode electrolyte interface, but this statement does no seem correct.

Author Response

(The authors gave the same response as above.)
